# Prior Convictions: Black-Box Adversarial Attacks with Bandits and Priors

**Andrew Ilyas**,[*] **Logan Engstrom**,[*] **Aleksander Mądry**
{ailyas, engstrom, madry}@mit.edu
MIT CSAIL

## Abstract

We study the problem of generating adversarial examples in a black-box setting in which only loss-oracle access to a model is available. We introduce a framework that conceptually unifies much of the existing work on black-box attacks, and we demonstrate that the current state-of-the-art methods are optimal in a natural sense. Despite this optimality, we show how to improve black-box attacks by bringing a new element into the problem: gradient priors. We give a bandit optimization-based algorithm that allows us to seamlessly integrate any such priors, and we explicitly identify and incorporate two examples. The resulting methods use two to four times fewer queries and fail two to five times less than the current state-of-the-art. [1]

## 1 Introduction

Recent research has shown that neural networks exhibit significant vulnerability to adversarial examples, or slightly perturbed inputs designed to fool the network prediction. This vulnerability is present in a wide range of settings, from situations in which inputs are fed directly to classifiers (Szegedy et al., 2013; Carlini et al., 2016) to highly variable real-world environments (Kurakin et al., 2016; Athalye et al., 2017). Researchers have developed a host of methods to construct such attacks (Goodfellow et al., 2014; Moosavi-Dezfooli et al., 2015; Carlini & Wagner, 2017; Madry et al., 2017), most of which correspond to first order (i.e., gradient based) methods. These attacks turn out to be highly effective: in many cases, only a few gradient steps suffice to construct an adversarial perturbation.

A significant shortcoming of many of these attacks, however, is that they fundamentally rely on the *white-box* threat model. That is, they crucially require direct access to the gradient of the classification loss of the attacked network. In many real-world situations, expecting this kind of complete access is not realistic. In such settings, an attacker can only issue classification queries to the targeted network, which corresponds to a more restrictive *black box* threat model.

Recent work (Chen et al., 2017; Bhagoji et al., 2017; Ilyas et al., 2017) provides a number of attacks for this threat model. Chen et al. (2017) show how to use a basic primitive of zeroth order optimization, the finite difference method, to estimate the gradient from classification queries and then use it (in addition to a number of optimizations) to mount a gradient based attack. The method indeed successfully constructs adversarial perturbations. It comes, however, at the cost of introducing a significant overhead in terms of the number of queries needed. For instance, attacking an ImageNet (Russakovsky et al., 2015) classifier requires hundreds of thousands of queries. Subsequent work (Ilyas et al., 2017) improves this dependence significantly, but still falls short of fully mitigating this issue (see Section 4.1 for a more detailed analysis).

### 1.1 Our contributions

We revisit zeroth-order optimization in the context of adversarial example generation, both from an empirical and theoretical perspective. We propose a new approach for generating black-box adversarial examples, using bandit optimization in order to exploit prior information about the gradient, which we show is necessary to break through the optimality of current methods. We

---

[1] The code for reproducing our work is available at `https://git.io/fAjOJ`.

evaluate our approach on the task of generating black-box adversarial examples, where the methods obtained from integrating two example priors significantly outperform state-of-the-art approaches.

Concretely, in this work:

1. We formalize the gradient estimation problem as the central problem in the context of query-efficient black-box attacks. We then show how the resulting framework unifies the previous attack methodology. We prove that the least squares method, a classic primitive in signal processing, not only constitutes an optimal solution to the general gradient estimation problem but also is essentially equivalent to the current-best black-box attack methods.

2. We demonstrate that, despite this seeming optimality of these methods, we can still improve upon them by exploiting an aspect of the problem that has been not considered previously: the priors we have on the distribution of the gradient. We identify two example classes of such priors, and show that they indeed lead to better predictors of the gradient.

3. Finally, we develop a bandit optimization framework for generating black-box adversarial examples which allows for the seamless integration of priors. To demonstrate its effectiveness, we show that leveraging the two aforementioned priors yields black-box attacks that are 2-5 times more query efficient and less failure-prone than the state of the art.

Table 1: Summary of effectiveness of $\ell_2$ and $\ell_\infty$ ImageNet attacks on Inception v3 using NES, bandits with time prior (Bandits$_T$), and bandits with time and data-dependent priors (Bandits$_{TD}$). Note that in the first column, the average number of queries is calculated only over successful attacks, and we enforce a query limit of 10,000 queries. For purposes of direct comparison, the last column calculates the average number of queries used for only the images that NES (previous SOTA) was successful on. Our most powerful attack uses 2-4 times fewer queries, and fails 2-5 times less often.

| Attack | Avg. Queries | | Failure Rate | | Queries on NES Success | |
|---|---|---|---|---|---|---|
| | $\ell_\infty$ | $\ell_2$ | $\ell_\infty$ | $\ell_2$ | $\ell_\infty$ | $\ell_2$ |
| NES | 1735 | 2938 | 22.2% | 34.4% | 1735 | 2938 |
| Bandits$_T$ | 1781 | 2690 | 11.6% | 30.4% | 1214 | 2421 |
| **Bandits$_{TD}$** | **1117** | **1858** | **4.6%** | **15.5%** | **703** | **999** |

## 2    BLACK-BOX ATTACKS AND THE GRADIENT ESTIMATION PROBLEM

Adversarial examples are natural inputs to a machine learning system that have been carefully perturbed in order to induce misbehaviour of the system, under a constraint on the magnitude of the pertubation (under some metric). For image classifiers, this misbehaviour can be either classification as a specific class other than the original one (the targeted attack) or misclassification (the untargeted attack). For simplicity and to make the presentation of the overarching framework focused, in this paper we restrict our attention to the untargeted case. Both our algorithms and the whole framework can be, however, easily adapted to the targeted setting. Also, we consider the most standard threat model in which adversarial perturbations must have $\ell_p$-norm, for some fixed $p$, less than some $\epsilon_p$.

### 2.1    FIRST-ORDER ADVERSARIAL ATTACKS

Suppose that we have some classifier $C(x)$ with a corresponding classification loss function $L(x, y)$, where $x$ is some input and $y$ its corresponding label. In order to generate a misclassified input from some input-label pair $(x, y)$, we want to find an adversarial example $x'$ which maximizes $L(x', y)$ but still remains $\epsilon_p$-close to the original input. We can thus formulate our adversarial attack problem as the following constrained optimization task:

$$x' = \underset{x' : \|x' - x\|_p \leq \epsilon_p}{\arg\max} \; L(x', y)$$

First order methods tend to be very successful at solving the problem despite its non-convexity (Goodfellow et al., 2014; Carlini & Wagner, 2017; Madry et al., 2017). A first order method used as the backbone of some of the most powerful white-box adversarial attacks for $\ell_p$ bounded adversaries is

*projected gradient descent (PGD).* This iterative method, given some input $x$ and its correct label $y$, computes a perturbed input $x_k$ by applying $k$ steps of the following update (with $x_0 = x$)

$$x_l = \Pi_{B_p(x,\epsilon)}(x_{l-1} + \eta s_l) \qquad\qquad \text{with } s_l = \Pi_{\partial B_p(0,1)} \nabla_x L(x_{l-1}, y) \qquad (1)$$

Here, $\Pi_S$ is the projection onto the set $S$, $B_p(x', \varepsilon')$ is the $\ell_p$ ball of radius $\varepsilon'$ around $x'$, $\eta$ is the step size, and $\partial U$ is the boundary of a set $U$. Also, as is standard in continuous optimization, we make $s_l$ be the projection of the gradient $\nabla_x L(x_{l-1}, y)$ at $x_{l-1}$ onto the unit $\ell_p$ ball. This way we ensure that $s_l$ corresponds to the unit $\ell_p$-norm vector that has the largest inner product with $\nabla_x L(x_{l-1}, y)$. (Note that, in the case of the $\ell_2$-norm, $s_l$ is simply the normalized gradient but in the case of, e.g., the $\ell_\infty$-norm, $s_l$ corresponds to the sign vector, $\text{sgn}(\nabla_x L(x_{l-1}, y))$ of the gradient.)

So, intuitively, the PGD update perturbs the input in the direction that (locally) increases the loss the most. Observe that due to the projection in (1), $x_k$ is always a valid perturbation of $x$, as desired.

## 2.2 BLACK-BOX ADVERSARIAL ATTACKS

The projected gradient descent (PGD) method described above is designed to be used in the context of so-called *white-box* attacks. That is, in the setting where the adversary has full access to the gradient $\nabla_x L(x, y)$ of the loss function of the attacked model. In many practical scenarios, however, this kind of access is not available—in the corresponding, more realistic *black-box* setting, the adversary has only access to an oracle that returns for a given input $(x, y)$, only the value of the loss $L(x, y)$.

One might expect that PGD is thus not useful in such black-box setting. It turns out, however, that this intuition is incorrect. Specifically, one can still *estimate* the gradient using only such value queries. (In fact, this kind of estimator is the backbone of so-called zeroth-order optimization frameworks (Spall, 2005).) The most canonical primitive in this context is the *finite difference method*. This method estimates the *directional* derivative $D_v f(x) = \langle \nabla_x f(x), v \rangle$ of some function $f$ at a point $x$ in the direction of a vector $v$ as

$$D_v f(x) = \langle \nabla_x f(x), v \rangle \approx (f(x + \delta v) - f(x))/\delta. \qquad (2)$$

Here, the step size $\delta > 0$ governs the quality of the gradient estimate. Smaller $\delta$ gives more accurate estimates but also decreases reliability, due to precision and noise issues. Consequently, in practice, $\delta$ is a tunable parameter. Now, we can just use finite differences to construct an estimate of the gradient. To this end, one can find the $d$ components of the gradient by estimating the inner products of the gradient with all the standard basis vectors $e_1, \ldots, e_d$:

$$\widehat{\nabla}_x L(x, y) = \sum_{k=1}^{d} e_k \left( L(x + \delta e_k, y) - L(x, y) \right)/\delta \approx \sum_{k=1}^{d} e_k \langle \nabla_x L(x, y), e_k \rangle \qquad (3)$$

We can then easily implement the PGD attack (c.f. (1)) using this estimator:

$$x_l = \Pi_{B_p(x,\epsilon)}(x_{l-1} + \eta \widehat{s}_l) \qquad\qquad \text{with } \widehat{s}_l = \Pi_{\partial B_p(0,1)} \widehat{\nabla}_x L(x_{l-1}, y) \qquad (4)$$

Indeed, Chen et al. (2017) were the first to use finite differences methods in this basic form to power PGD–based adversarial attack in the black-box setting. This basic attack was shown to be successful but, since its query complexity is proportional to the dimension, its resulting query complexity was prohibitively large. For example, the Inception v3 (Szegedy et al., 2015) classifier on the ImageNet dataset has dimensionality d=268,203 and thus this method would require 268,204 queries. (It is worth noting, however, that Chen et al. (2017) developed additional methods to, at least partially, reduce this query complexity.)

## 2.3 BLACK-BOX ATTACKS WITH IMPERFECT GRADIENT ESTIMATORS

In the light of the above discussion, one can wonder if the algorithm (4) can be made more query-efficient. A natural idea here would be to avoid fully estimating the gradient and rely instead only on its *imperfect* estimators. This gives rise to the following question: *How accurate of an gradient estimate is necessary to execute a successful PGD attack?*

We examine this question first in the simplest possible setting: one in which we only take a *single* PGD step (i.e., the case of $k = 1$). Previous work (Goodfellow et al., 2014) indicates that such an

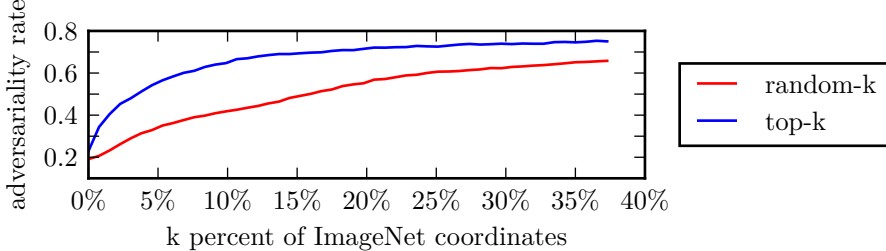

Figure 1: The fraction of correctly estimated coordinates of $\text{sgn}(\nabla_x L(x, y))$ required to successfully execute the single-step PGD (also known as FGSM) attack, with $\epsilon = 0.05$. In the experiment, for each $k$, the top $k$ percent – chosen either by magnitude (`top-k`) or randomly (`random-k`) – of the signs of the coordinates are set correctly, and the rest are set to $+1$ or $-1$ at random. The adversariality rate is the portion of 1,000 random ImageNet images misclassified after one FGSM step. For example, estimating only 20% of coordinates correctly leads to misclassification for $> 60\%$ of images.

attack can already be quite powerful. So, we study how the effectiveness of this attack varies with gradient estimator accuracy. Our experiments, shown in Figure 1, suggest that it is feasible to generate adversarial examples without estimating correctly even most of the coordinates of the gradient. For example, in the context of $\ell_\infty$ attacks, setting a randomly selected 20% of the coordinates in the gradient to match the true gradient (and making the remaining coordinates have random sign) is sufficient to fool the classifier on more than 60% images with single-step PGD. Our experiments thus demonstrate that an adversary is likely to be able to cause a misclassification by performing the iterated PGD attack, even when driven by a gradient estimate that is largely imperfect.

## 2.4 THE GRADIENT ESTIMATION PROBLEM

The above discussion makes it clear that successful attacks do not require a perfect gradient estimation, provided this estimate is suitably constructed. It is still unclear, however, how to efficiently find this kind of imperfect but helpful estimator. Continuous optimization methodology suggests that the key characteristic needed from our estimator is for it to have a sufficiently large inner product with the actual gradient. We thus capture this challenge as the following *gradient estimation problem*:

**Definition 1** (Gradient estimation problem). *For an input/label pair $(x, y)$ and a loss function L, let $g^* = \nabla_x L(x, y)$ be the gradient of L at $(x, y)$. Then the goal of the gradient estimation problem is to find a unit vector $\widehat{g}$ maximizing the inner product*

$$\mathbb{E}\left[\widehat{g}^T g^*\right], \tag{5}$$

*from a limited number of (possibly adaptive) function value queries $L(x', y')$. (The expectation here is taken over the randomness of the estimation algorithm.)*

One useful perspective on the above gradient estimation problem stems from casting the recovery of $g^*$ in (5) as an underdetermined vector estimation task. That is, one can view each execution of the finite difference method (see (2)) as computing an inner product query in which we obtain the value of the inner product of $g^*$ and some chosen direction vector $A_i$. Now, if we execute $k$ such queries, and $k < d$ (which is the regime we are interested in), the information acquired in this process can be expressed as the following (underdetermined) linear regression problem $Ag^* = y$, where the rows of the matrix $A$ correspond to the queries $A_1, \ldots, A_k$ and the entries of the vector $y$ gives us the corresponding inner product values.

**Relation to compressive sensing.** The view of the gradient estimation problem we developed bears striking similarity to the compressive sensing setting (Foucart & Rauhut, 2013). Thus one might wonder if the toolkit of that area could be applied here. Compressive sensing crucially requires, however, certain sparsity structure in the estimated signal (here, in the gradient $g^*$) and, to our knowledge, the loss gradients do not exhibit such a structure. (We discuss this further in Appendix B.)

**The least squares method.** In light of this, we turn our attention to another classical signal-processing method: norm-minimizing $\ell_2$ least squares estimation. This method approaches the estimation

problem posed in (5) by casting it as an undetermined linear regression problem of the form $Ag^* = b$, where we can choose the matrix $A$ (the rows of $A$ correspond to inner product queries with $g^*$). Then, it obtains the solution $\widehat{g}$ to the regression problem by solving:

$$\min_{\widehat{g}} \|\widehat{g}\|_2 \qquad \text{s.t. } A\widehat{g} = y. \tag{6}$$

A reasonable choice for $A$ (via Johnson & Lindenstrauss (1984) and related results) is the distance-preserving random Gaussian projection matrix, i.e. $A_{ij}$ normally distributed.

The resulting algorithm turns out to yield solutions that are approximately those given by Natural Evolution Strategies (NES), which (Ilyas et al., 2017) previously applied to black-box attacks. In particular, in Appendix A, we prove the following theorem.

**Theorem 1** (NES and Least Squares equivalence). *Let $\hat{x}_{NES}$ be the Gaussian $k$-query NES estimator of a $d$-dimensional gradient $\boldsymbol{g}$ and let $\hat{x}_{LSQ}$ be the minimal-norm $k$-query least-squares estimator of $\boldsymbol{g}$. For any $p > 0$, with probability at least $1 - p$ we have that*

$$\langle \hat{x}_{LSQ}, \boldsymbol{g} \rangle - \langle \hat{x}_{NES}, \boldsymbol{g} \rangle \leq O\left( \sqrt{(k/d) \cdot \log^3\left((k/p)\right)} \right) \|g\|^2.$$

Note that when we work in the underdetermined setting, i.e., when $k \ll d$ (which is the setting we are interested in), the right hand side bound becomes vanishingly small. Thus, the equivalence indeed holds. In fact, using the precise statement (given and proved in Appendix A), we can show that Theorem 1 provides us with a non-vacuous equivalence bound. Further, it turns out that one can exploit this equivalence to prove that the algorithm proposed in Ilyas et al. (2017) is not only natural but optimal, as the least-squares estimate is an information-theoretically optimal gradient estimate in the regime where $k = d$, and an error-minimizing estimator in the regime where $k << d$.

**Theorem 2** (Least-squares optimality (Proof in Appendix A)). *For a linear regression problem $y = A\boldsymbol{g}$ with known $A$ and $y$, unknown $\boldsymbol{g}$, and isotropic Gaussian errors, the least-squares estimator is finite-sample efficient, i.e. the minimum-variance unbiased (MVU) estimator of the latent vector $\boldsymbol{g}$.*

**Theorem 3** (Least-squares optimality (Proof in Meir (1994))). *In the underdetermined setting, i.e. when $k << d$, the minimum-norm least squares estimate ($\hat{x}_{LSQ}$ in Theorem 1) is the minimum-variance (and thus minimum-error, since bias is fixed) estimator with no empirical loss.*

## 3   BLACK-BOX ADVERSARIAL ATTACKS WITH PRIORS

The optimality of least squares strongly suggests that we have reached the limit of query-efficiency of black-box adversarial attacks. But is this really the case? Surprisingly, we show that an improvement *is* still possible. The key observation is that the optimality we established of least-squares (and by Theorem 1, the NES approach in (Ilyas et al., 2017)) holds only for the most basic setting of the gradient estimation problem, a setting where we assume that the target gradient is a truly arbitrary and completely unknown vector.

However, in the context we care about this assumption does not hold – there is actually plenty of prior knowledge about the gradient available. Firstly, the input with respect to which we compute the gradient is not arbitrary and exhibits locally predictable structure which is consequently reflected in the gradient. Secondly, when performing iterative gradient attacks (e.g. PGD), the gradients used in successive iterations are likely to be heavily correlated.

The above observations motivate our focus on *prior information* as an integral element of the gradient estimation problem. Specifically, we enhance Definition 1 by making its objective

$$\mathbb{E}\left[\widehat{g}^T g^* \,\middle|\, I\right], \text{ where } I \text{ is prior information available to us.} \tag{7}$$

This change in perspective gives rise to two important questions: *does there exist prior information that can be useful to us?*, and *does there exist an algorithmic way to exploit this information?* We show that the answer to both of these questions is affirmative.

### 3.1   GRADIENT PRIORS

Consider a gradient $\nabla_x L(x, y)$ of the loss function corresponding to some input $(x, y)$. Does there exist some kind of prior that can be extracted from the dataset $\{x_i\}$, in general, and the input $(x, y)$

in particular, that can be used as a predictor of the gradient? We demonstrate that it is indeed the case, and give two example classes of such priors.

**Time-dependent priors.** The first class of priors we consider are time-dependent priors, a standard example of which is what we refer to as the "multi-step prior." We find that along the trajectory taken by estimated gradients, successive gradients are in fact heavily correlated. We show this empirically by taking steps along the optimization path generated by running the NES estimator at each point, and plotting the normalized inner product (cosine similarity) between successive gradients, given by

$$\frac{\langle \nabla_x L(x_t, y), \nabla_x L(x_{t+1}, y) \rangle}{||\nabla_x L(x_t, y)||_2 ||\nabla_x L(x_{t+1}, y)||_2} \qquad t \in \{1 \dots T - 1\}. \tag{8}$$

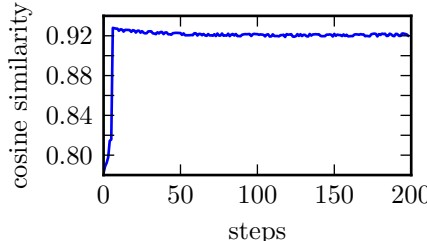
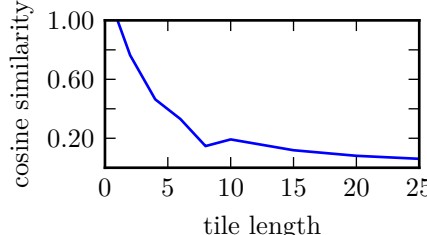

Figure 2: Cosine similarity between the gradients at the current and previous steps along the optimization trajectory of NES PGD attacks, averaged over 1000 random ImageNet images.

Figure 3: Cosine similarity of "tiled" image gradient with original image gradient versus the length of the square tiles, averaged over 5,000 randomly selected ImageNet images.

Figure 2 demonstrates that there indeed is a non-trivial correlation between successive gradients—typically, the gradients of successive steps (using step size from Ilyas et al. (2017)) have a cosine similarity of about 0.9. Successive gradients continue to correlate at higher step sizes: Appendix B shows that the trend continues even at step size 4.0 (a typical value for the *total* perturbation bound $\varepsilon$). This indicates that there indeed is a potential gain from incorporating this correlation into our iterative optimization. To utilize this gain, we intend to use the gradients at time $t - 1$ as a prior for the gradient at time $t$, where both the prior and the gradient estimate itself evolve over iterations.

**Data-dependent priors.** We find that the time-dependent prior discussed above is not the only type of prior one can exploit here. Namely, we can also use the structure of the inputs themselves to reduce query complexity (in fact, the existence of such data-dependent priors is what makes machine learning successful in the first place).

In the case of image classification, a simple and heavily exploited example of such a prior stems from the fact that images tend to exhibit a spatially local similarity (i.e. pixels that are close together tend to be similar). We find that this similarity also extends to the gradients: specifically, whenever two coordinates $(i, j)$ and $(k, l)$ of $\nabla_x L(x, y)$ are close, we expect $\nabla_x L(x, y)_{ij} \approx \nabla_x L(x, y)_{kl}$ too. To corroborate and quantify this phenomenon, we compare $\nabla_x L(x, y)$ with an average-pooled, or "tiled", version (with "tile length" $k$) of the same signal. An example of such an average-blurred gradient can be seen in Appendix B. More concretely, we apply to the gradient the mean pooling operation with kernel size $(k, k, 1)$ and stride $(k, k, 1)$, then upscale the spatial dimensions by $k$. We then measure the cosine similarity between the average-blurred gradient and the gradient itself. Our results, shown in Figure 3, demonstrate that the gradients of images are locally similar enough to allow for average-blurred gradients to maintain relatively high cosine similarity with the actual gradients, even when the tiles are large. Our results suggest that we can reduce the dimensionality of our problem by a factor of $k^2$ (for reasonably large $k$) and still estimate a vector pointing close to the same direction as the original gradient. This factor, as we show later, leads to significantly improved black-box adversarial attack performance.

### 3.2 A FRAMEWORK FOR GRADIENT ESTIMATION WITH PRIORS

Given the availability of these informative gradient priors, we now need a framework that enables us to easily incorporate these priors into our construction of black-box adversarial attacks. Our

proposed method builds on the framework of *bandit optimization*, a fundamental tool in online convex optimization Hazan (2016). In the bandit optimization framework, an agent plays a game that consists of a sequence of rounds. In round $t$, the agent must choose a valid action, and then by playing the action incurs a loss given by a loss function $\ell_t(\cdot)$ that is unknown to the agent. After playing the action, he/she only learns the loss that the chosen action incurs; the loss function is specific to the round $t$ and may change arbitrarily between rounds. The goal of the agent is to minimize the average loss incurred over all rounds, and the success of the agent is usually quantified by comparing the total loss incurred to that of the *best expert in hindsight* (the best single-action policy). By the nature of this formulation, the rounds of this game can not be treated as independent — to perform well, the agent needs to keep track of some latent record that aggregates information learned over a sequence of rounds. This latent record usually takes a form of a vector $v_t$ that is constrained to a specified (convex) set $\mathcal{K}$. As we will see, this aspect of the bandit optimization framework will provide us with a convenient way to incorporate prior information into our gradient prediction.

**An overview of gradient estimation with bandits.** We can cast the gradient estimation problem as an bandit optimization problem in a fairly direct manner. Specifically, we let the action at each round $t$ be a gradient estimate $g_t$ (based on our latent vector $v_t$), and the loss $\ell_t$ correspond to the (negative) inner product between this prediction and the actual gradient. Note that we will never have a direct access to this loss function $\ell_t$ but we are able to evaluate its value on a particular prediction vector $g_t$ via the finite differences method (2) (which is all that the bandits optimization framework requires us to be able to do).

Just as this choice of the loss function $\ell_t$ allows us to quantify performance on the gradient estimation problem, the latent vector $v_t$ will allow us to algorithmically incorporate prior information into our predictions. Looking at the two example priors we consider, the time-dependent prior will be reflected by carrying over the latent vector between the gradient estimations at different points. Data-dependent priors will be captured by enforcing that our latent vector has a particular structure. For the specific prior we quantify in the preceding section (data-dependent prior for images), we will simply reduce the dimensionality of the latent vector via average-pooling ("tiling"), removing the need for extra queries to discern components of the gradient that are spatially close.

### 3.3 IMPLEMENTING GRADIENT ESTIMATION IN THE BANDIT FRAMEWORK

We now describe our bandit framework for adversarial example generation in more detail. Note that the algorithm is general and can be used to construct black-box adversarial examples where the perturbation is constrained to any convex set ($\ell_p$-norm constraints being a special case). We discuss the algorithm in its general form, and then provide versions explicitly applied to the $\ell_2$ and $\ell_\infty$ cases.

As previously mentioned, the latent vector $v_t \in \mathcal{K}$ serves as a prior on the gradient for the corresponding round $t$ – in fact, we make our prediction $g_t$ be exactly $v_t$ projected onto the appropriate space, and thus we set $\mathcal{K}$ to be an extension of the space of valid adversarial perturbations (e.g. $\mathbb{R}^n$ for $\ell_2$ examples, $[-1, 1]^n$ for $\ell_\infty$ examples). Our loss function $\ell_t$ is defined as

$$\ell_t(g) = -\langle \nabla L(x, y), \frac{g}{||g||}\rangle, \tag{9}$$

for a given gradient estimate $g$, where we access this inner product via finite differences. Here, $L(x, y)$ is the classification loss on an image $x$ with true class $y$.

The crucial element of our algorithm will thus be the method of updating the latent vector $v_t$. We will adapt here the canonical "reduction from bandit information" (Hazan, 2016). Specifically, our update procedure is parametrized by an estimator $\Delta_t$ of the gradient $\nabla_v \ell_t(v)$, and a first-order update step $\mathcal{A}$ ($\mathcal{K} \times \mathbb{R}^{\dim(\mathcal{K})} \to \mathcal{K}$), which maps the latent vector $v_t$ and the estimated gradient of $\ell_t$ with respect to $v_t$ (which we denote $\Delta_t$) to a new latent vector $v_{t+1}$. The resulting general algorithm is presented as Algorithm 1.

In our setting, we make the estimator $\Delta$ of the gradient $-\nabla_v \langle \nabla L(x, y), v \rangle$ of the loss $\ell$ be the standard spherical gradient estimator (see Hazan (2016)). We take a two-query estimate of the expectation, and employ antithetic sampling which results in the estimate being computed as

$$\Delta = \frac{\ell(v + \delta \boldsymbol{u}) - \ell(v - \delta \boldsymbol{u})}{\delta} \boldsymbol{u}, \tag{10}$$

---

**Algorithm 1** Gradient Estimation with Bandit Optimization

---

1: **procedure** BANDIT-OPT-LOSS-GRAD-EST$(x, y_{init})$
2:      $v_0 \leftarrow \mathcal{A}(\phi)$
3:      **for** each round $t = 1, \ldots, T$ **do**
4:          // Our loss in round $t$ is $\ell_t(g_t) = -\langle \nabla_x L(x, y_{init}), g_t \rangle$
5:          $g_t \leftarrow v_{t-1}$
6:          $\Delta_t \leftarrow$ GRAD-EST$(x, y_{init}, v_{t-1})$ // Estimated Gradient of $\ell_t$
7:          $v_t \leftarrow \mathcal{A}(v_{t-1}, \Delta_t)$
8:      $g \leftarrow v_T$
9:      **return** $\Pi_{\partial \mathcal{K}}[g]$

---

where $\boldsymbol{u}$ is a Gaussian vector sampled from $\mathcal{N}(0, \frac{1}{d}I)$. The resulting algorithm for calculating the gradient estimate given the current latent vector $v$, input $x$ and the initial label $y$ is Algorithm 2.

---

**Algorithm 2** Single-query spherical estimate of $\nabla_v \langle \nabla L(x, y), v \rangle$

---

1: **procedure** GRAD-EST$(x, y, v)$
2:      $u \leftarrow \mathcal{N}(0, \frac{1}{d}I)$ // Query vector
3:      $\{q_1, q_2\} \leftarrow \{v + \delta \boldsymbol{u}, v - \delta \boldsymbol{u}\}$ // Antithetic samples
4:      $\ell_t(q_1) = -\langle \nabla L(x, y), q_1 \rangle \approx \frac{L(x,y) - L(x + \epsilon \cdot q_1, y)}{\epsilon}$ // Gradient estimation loss at $q_1$
5:      $\ell_t(q_2) = -\langle \nabla L(x, y), q_2 \rangle \approx \frac{L(x,y) - L(x + \epsilon \cdot q_2, y)}{\epsilon}$ // Gradient estimation loss at $q_2$
6:      $\Delta \leftarrow \frac{\ell_t(q_1) - \ell_t(q_2)}{\delta} \boldsymbol{u} = \frac{L(x + \epsilon q_2, y) - L(x + \epsilon q_1, y)}{\delta \epsilon} \boldsymbol{u}$
7:      // Note that due to cancellations we can actually evaluate $\Delta$ with only two queries to $L$
8:      **return** $\Delta$

---

A crucial point here is that the above gradient estimator $\Delta_t$ parameterizing the bandit reduction has no direct relation to the "gradient estimation problem" as defined in Section 2.4. It is simply a general mechanism by which we can update the latent vector $v_t$ in bandit optimization. It is the actions $g_t$ (equal to $v_t$) which provide proposed solutions to the gradient estimation problem from Section 2.4.

The choice of the update rule $\mathcal{A}$ tends to be natural once the convex set $\mathcal{K}$ is known. For $\mathcal{K} = \mathbb{R}^n$, we can simply use gradient ascent:

$$v_t = \mathcal{A}(v_{t-1}, \Delta_t) := v_{t-1} + \eta \cdot \Delta_t \tag{11}$$

and the exponentiated gradients (EG) update when the constraint is an $\ell_\infty$ bound (i.e. $\mathcal{K} = [-1, 1]^n$):

$$p_{t-1} = \frac{1}{2}(v_{t-1} + 1)$$

$$p_t = \mathcal{A}(g_{t-1}, \Delta_t) := \frac{1}{Z} p_{t-1} \exp(\eta \cdot \Delta_t) \quad \text{s.t. } Z = p_{t-1} \exp(\eta \cdot \Delta_t) + (1 - p_{t-1}) \exp(-\eta \cdot \Delta_t)$$

$$v_t = 2p_t - 1$$

Finally, in order to translate our gradient estimation algorithm into an efficient method for constructing black-box adversarial examples, we interleave our iterative gradient estimation algorithm with an iterative update of the image itself, using the boundary projection of $g_t$ in place of the gradient (c.f. (1)). This results in a general, efficient, prior-exploiting algorithm for constructing black-box adversarial examples. The resulting algorithm in the $\ell_2$-constrained case is shown in Algorithm 3.

## 4 EXPERIMENTS AND EVALUATION

We evaluate our bandit approach described in Section 3 and the natural evolutionary strategies (NES) approach of Ilyas et al. (2017) on their effectiveness in generating untargeted adversarial examples. We consider both the $\ell_2$ and $\ell_\infty$ threat models on the ImageNet (Russakovsky et al., 2015) dataset, in terms of success rate and query complexity. We further investigate loss and gradient estimate quality over the optimization trajectory in each method. To show the method extends to other datasets,

---

**Algorithm 3** Adversarial Example Generation with Bandit Optimization for $\ell_2$ norm perturbations

---

1: **procedure** ADVERSARIAL-BANDIT-L2($x_{init}, y_{init}$)
2:     // $C(\cdot)$ returns top class
3:     $v_0 \leftarrow \mathbf{0}_{1 \times d}$ // If data prior, $d < \dim(x)$; $v_t$ ($\Delta_t$) up (down)-sampled before (after) line 8
4:     $x_0 \leftarrow x_{init}$ // Adversarial image to be constructed
5:     **while** $C(x) = y_{init}$ **do**
6:         $g_t \leftarrow v_{t-1}$
7:         $x_t \leftarrow x_{t-1} + h \cdot \frac{g_t}{||g_t||_2}$ // Boundary projection $\frac{g}{||g||}$ standard PGD: c.f. (Rigollet, 2015)
8:         $\Delta_t \leftarrow$ GRAD-EST($x_{t-1}, y_{init}, v_{t-1}$) // Estimated Gradient of $\ell_t$
9:         $v_t \leftarrow v_{t-1} + \eta \cdot \Delta_t$
10:        $t \leftarrow t + 1$
      **return** $x_{t-1}$

---

we also compare to NES in the CIFAR-$\ell_\infty$ threat model; in all threat models, we show results on Inception-v3, Resnet-50, and VGG16 classifiers.

In evaluating our approach, we test both the bandit approach with time prior (Bandits$_T$), and our bandit approach with the given examples of both the data and time priors (Bandits$_{TD}$). We use 10,000 and 1,000 randomly selected images (scaled to $[0, 1]$) to evaluate all approaches on ImageNet and CIFAR-10 respectively. For NES, Bandits$_T$, and Bandits$_{TD}$ we found hyperparameters (given in Appendix C, along with the experimental parameters) via grid search.

### 4.1 RESULTS

For ImageNet, we record the effectiveness of the different approaches in both threat models in Table 1 ($\ell_2$ and $\ell_\infty$ perturbation constraints), where we show the attack success rate and the mean number of queries (of the successful attacks) needed to generate an adversarial example for the Inception-v3 classifier (results for other classifiers in Appendix F). For all attacks, we limit the attacker to at most 10,000 oracle queries. As shown in Table 1, our bandits framework with both data-dependent and time prior (Bandits$_{TD}$), is six and three times less failure-prone than the previous state of the art (NES (Ilyas et al., 2017)) in the $\ell_\infty$ and $\ell_2$ settings, respectively. Despite the higher success rate, our method actually uses around half as many queries as NES. In particular, when restricted to the inputs on which NES is successful in generating adversarial examples, our attacks are 2.5 and 5 times as query-efficient for the $\ell_\infty$ and $\ell_2$ settings, respectively. In Appendix G, we also compare against the AutoZOOM method of Tu et al. (2018), where we show that our Bandits$_{TD}$ method at a higher $100\%$ success rate is over *6 times* as query-efficient. Finally, we also have similar results for CIFAR-10 under the $\ell_\infty$ threat model, which can be found in Appendix E.

We also further quantify the performance of our methods in terms of black-box attacks, and gradient estimation. Specifically, we first measure average queries per success after reaching a certain success rate (Figure 4a), which indicates the dependence of the query count on the desired success rate. The data shows that for any fixed success rate, our methods are more query-efficient than NES, and (due to the exponential trend) suggest that the difference may be amplified for higher success rates. We then plot the loss of the classifier over time (averaged over all images), and performance on the gradient estimation problem for both $\ell_\infty$ and $\ell_2$ cases (which, crucially, corresponds directly to the expectation we maximize in (7). We show these three plots for $\ell_\infty$ in Figure 4, and show the results for $\ell_2$ (which are extremely similar) in Appendix D, along with CDFs showing the success of each method as a function of the query limit. We find that on every metric in both threat models, our methods strictly dominate NES in terms of performance.

## 5 RELATED WORK

All known techniques for generating adversarial examples in the black-box setting so far rely on either iterative optimization schemes (our focus) or so-called substitute networks and transferability.

In the first line of work, algorithms use queries to gradually perturb a given input to maximize a corresponding loss, causing misclassification. Nelson et al. (2012) presented the first such iterative attack on a special class of binary classifiers. Later, Xu et al. (2016) gave an algorithm for fooling a

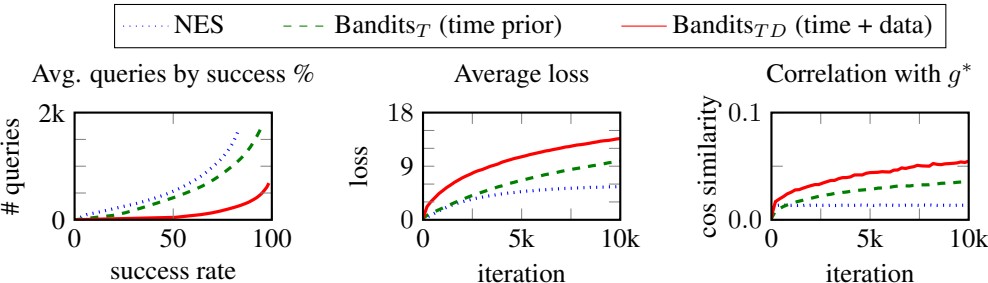

Figure 4: **(left)** Average number of queries per successful image as a function of the number of total successful images; at any desired success rate, our methods use significantly less queries per successful image than NES, and the trend suggests that this gap increases with the desired success rate. **(center)** The loss over time, averaged over all images; **(right)** The correlation of the latent vector with the true gradient $g$, which is precisely the gradient estimation objective we define.

real-world system with black-box attacks. Specifically, they fool PDF document malware classifier by using a genetic algorithms-based attack. Soon after, Narodytska & Kasiviswanathan (2017) described the first black-box attack on deep neural networks; the algorithm uses a greedy search algorithm that selectively changes individual pixel values. Chen et al. (2017) were the first to design black-box attack based on finite-differences and gradient based optimization. The method uses coordinate descent to attack black-box neural networks, and introduces various optimizations to decrease sample complexity. Building on the work of Chen et al. (2017), Ilyas et al. (2017) designed a black-box attack strategy that also uses finite differences but via natural evolution strategies (NES) to estimate the gradients. They then used their algorithm as a primitive in attacks on more restricted threat models.

In a concurrent line of work, Papernot et al. (2017) introduce a method for attacking models with so-called substitute networks. Here, the attacker trains a model – called a substitute network – to mimic the target network's decisions (obtained with black-box queries) , then uses (white-box) adversarial examples for the substitute network to attack the original model. Adversarial examples generated with these methods Papernot et al. (2017); Liu et al. (2016) tend to transfer to a target MNIST or CIFAR classifier. We note, however, that for attacking single inputs, the overall query efficiency of this type of methods tends to be worse than that of the gradient estimation based ones. Substitute models are also thus far unable to make targeted black-box adversarial examples.

## 6 CONCLUSION

We develop a new, unifying perspective on black-box adversarial attacks. This perspective casts the construction of such attacks as a gradient estimation problem. We prove that a standard least-squares estimator both captures the existing state-of-the-art approaches to black-box adversarial attacks, and actually is, in a certain natural sense, an optimal solution to the problem.

We then break the barrier posed by this optimality by considering a previously unexplored aspect of the problem: the fact that there exists plenty of extra prior information about the gradient that one can exploit to mount a successful adversarial attack. We identify two examples of such priors: a "time-dependent" prior that corresponds to similarity of the gradients evaluated at similar inputs, and a "data-dependent" prior derived from the latent structure present in the input space.

Finally, we develop a bandit optimization approach to black-box adversarial attacks that allows for a seamless integration of such priors. The resulting framework significantly outperforms state-of-the-art by a factor of two to six in terms of success rate and query efficiency. Our results thus open a new avenue towards finding priors for construction of even more efficient black-box adversarial attacks.

### ACKNOWLEDGMENTS

We thank Ludwig Schmidt for suggesting the connection between LSQ and NES. AM supported in part by NSF grants CCF-1553428 and CNS-1815221. LE supported in part by a Siebel Foundation Scholarship and IBM Watson AI grant. AI supported by an Analog Devices Fellowship.

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

## A  PROOFS

**Theorem 1** (NES and Least Squares equivalence). *Let $\hat{x}_{NES}$ be the Gaussian $k$-query NES estimator of a $d$-dimensional gradient $\boldsymbol{g}$ and let $\hat{x}_{LSQ}$ be the minimal-norm $k$-query least-squares estimator of $\boldsymbol{g}$. For any $p > 0$, with probability at least $1 - p$ we have that*

$$\langle \hat{x}_{LSQ}, \boldsymbol{g} \rangle - \langle \hat{x}_{NES}, \boldsymbol{g} \rangle \leq O\left( \sqrt{\frac{k}{d} \cdot \log^3\left(\frac{k}{p}\right)} \right) ||g||^2,$$

*and in particular,*

$$\langle \hat{x}_{LSQ}, \boldsymbol{g} \rangle - \langle \hat{x}_{NES}, \boldsymbol{g} \rangle \leq 8\sqrt{\frac{2k}{d} \cdot \log^3\left(\frac{2k+2}{p}\right)} \left(1 + \frac{\kappa}{\sqrt{d}}\right) ||g||^2$$

*with probability at least $1 - p$, where*

$$\kappa \leq 2\sqrt{\log\left(\frac{2k(k+1)}{p}\right)}.$$

*Proof.* Let us first recall our estimation setup. We have $k$ query vectors $\delta_i \in \mathbb{R}^d$ drawn from an i.i.d Gaussian distribution whose expected squared norm is one, i.e. $\delta_i \sim \mathcal{N}(0, \frac{1}{d}I)$, for each $1 \leq i \leq k$. Let the vector $\boldsymbol{y} \in \mathbb{R}^k$ denote the inner products of $\delta_i$s with the gradient, i.e.

$$y_i := \langle \delta_i, \boldsymbol{g} \rangle,$$

for each $1 \leq i \leq k$. We define the matrix $A$ to be a $k \times d$ matrix with the $\delta_i$s being its rows. That is, we have

$$A\boldsymbol{g} = \boldsymbol{y}.$$

Now, recall that the closed forms of the two estimators we are interested in are given by

$$\hat{x}_{NES} = A^T \boldsymbol{y} = A^T A \boldsymbol{g}$$
$$\hat{x}_{LSQ} = A^T (AA^T)^{-1} \boldsymbol{y} = A^T (AA^T)^{-1} A \boldsymbol{g},$$

which implies that

$$\langle \hat{x}_{NES}, \boldsymbol{g} \rangle = \boldsymbol{g}^T A^T A \boldsymbol{g}$$
$$\langle \hat{x}_{LSQ}, \boldsymbol{g} \rangle = \boldsymbol{g}^T A^T (AA^T)^{-1} A \boldsymbol{g}.$$

We can bound the difference between these two inner products as

$$\begin{aligned}
\langle \hat{x}_{LSQ}, \boldsymbol{g} \rangle - \langle \hat{x}_{NES}, \boldsymbol{g} \rangle &= \boldsymbol{g}^T A^T \left[ (AA^T)^{-1} - I \right] A\boldsymbol{g} \\
&\leq ||\boldsymbol{g}^T A^T|| \, ||(AA^T)^{-1} - I|| \, ||A\boldsymbol{g}|| \\
&\leq ||(AA^T)^{-1} - I|| \, ||A\boldsymbol{g}||^2 .
\end{aligned} \quad (12)$$

Now, to bound the first term in (12), observe that

$$(AA^T)^{-1} = \left(I - (I - AA^T)\right)^{-1} = \sum_{l=0}^{\infty} (I - AA^T)^l$$

and thus

$$I - (AA^T)^{-1} = \sum_{l=1}^{\infty} (AA^T - I)^l.$$

(Note that the first term in the above sum has been canceled out.) This gives us that

$$||I - (AA^T)^{-1}|| \leq \sum_{l=1}^{\infty} ||AA^T - I||^l$$

$$\leq \frac{\left|\left|AA^T - I\right|\right|}{1 - \left|\left|AA^T - I\right|\right|}$$

$$\leq 2\left|\left|AA^T - I\right|\right|,$$

as long as $\left|\left|AA^T - I\right|\right| \leq \frac{1}{2}$ (which, as we will see, is indeed the case with high probability).

Our goal thus becomes bounding $\left|\left|AA^T - I\right|\right| = \lambda_{max}(AA^T - I)$, where $\lambda_{max}(\cdot)$ denotes the largest (in absolute value) eigenvalue. Observe that $AA^T$ and $-I$ commute and are simultaneously diagonalizable. As a result, for any $1 \leq i \leq k$, we have that the $i$-th largest eigenvalue $\lambda_i(AA^T - I)$ of $AA^T - I$ can be written as

$$\lambda_i(AA^T - I) = \lambda_i(AA^T) + \lambda_i(-I)_i = \lambda_i(AA^T) - 1.$$

So, we need to bound

$$\lambda_{max}(AA^T - I) = \max\left\{\lambda_1(AA^T) - 1, 1 - \lambda_k(AA^T)\right\}$$

To this end, recall that $\mathbb{E}[AA^T] = I$ (since the rows of $A$ are sampled from the distribution $\mathcal{N}(0, \frac{1}{d}I)$), and thus, by the covariance estimation theorem of Gittens and Tropp Gittens & Tropp (2011) (see Corollary 7.2) (and union bounding over the two relevant events), we have that

$$\Pr(\lambda_{max}(AA^T - I) \geq \varepsilon) = \Pr(\lambda_1(AA^T) \geq 1 + \varepsilon \text{ or } \lambda_k(AA^T) \geq 1 - \varepsilon)$$

$$= \Pr(\lambda_1(AA^T) \geq \lambda_1(I) + \varepsilon \text{ or } \lambda_k(AA^T) \geq \lambda_k(I) - \varepsilon) \leq 2k \cdot \exp\left(-\frac{d\varepsilon^2}{32k}\right).$$

Setting

$$\varepsilon = \sqrt{\frac{32k \log(2(k+1)/p)}{d}},$$

ensuring that $\varepsilon \leq \frac{1}{2}$, gives us

$$\Pr\left(\lambda_{max}(AA^T) - 1 \geq \sqrt{\frac{32k \log(2(k+1)/p)}{d}}\right) \leq \frac{k}{k+1}p.$$

and thus

$$\left|\left|(AA^T)^{-1} - I\right|\right| \leq \sqrt{\frac{32k \log(2(k+1)/p)}{d}}, \tag{13}$$

with probability at least $1 - \frac{k}{k+1}p$.

To bound the second term in (12), we note that all the vectors $\delta_i$ are chosen independently of the vector $g$ and each other. So, if we consider the set $\{\hat{g}, \hat{\delta}_1, \ldots, \hat{\delta}_k\}$ of $k+1$ corresponding *normalized* directions, we have (see, e.g., (Gorban et al., 2016)) that the probability that any two of them have the (absolute value of) their inner product be larger than some $\varepsilon' = \sqrt{\frac{2\log(2(k+1)/p)}{d}}$ is at most

$$\exp\left\{-(k+1)^2 e^{-d(\varepsilon')^2/2}\right\} = \exp\left\{-2\frac{k+1}{p}\right\} \leq \frac{p}{2(k+1)}.$$

On the other hand, we note that each $\delta_i$ is a random vector sampled from the distribution $\mathcal{N}(0, \frac{1}{d}\boldsymbol{I}_d)$, so we have that (see, e.g., Lemma 1 in (Laurent & Massart, 2000)), for any $1 \leq i \leq k$ and any $\varepsilon'' > 0$,

$$\Pr\left(||\delta_i||^2 \geq 1 + \varepsilon''\right) \leq \exp\left\{-\frac{(\varepsilon'')^2 d}{4}\right\}.$$

Setting

$$\varepsilon'' = 2\sqrt{\frac{\log(2k(k+1)/p)}{d}}$$

yields

$$P\left(||\delta_i||^2 \geq 1 + 2\sqrt{\frac{\log(2(k+1)k/p)}{d}}\right) \leq \frac{p}{2k(k+1)}.$$

Applying these two bounds (and, again, union bounding over all the relevant events), we get that

$$||A\boldsymbol{g}||^2 = \sum_{i=1}^{k} (A\boldsymbol{g})_i^2$$

$$\leq d \cdot \left( \frac{2 \log \left( \frac{2(k+1)}{p} \right)}{d} \right) \left( 1 + 2 \sqrt{\frac{\log \left( \frac{2k(k+1)}{p} \right)}{d}} \right) ||g||^2$$

$$\leq 2 \log \left( \frac{2(k+1)}{p} \right) \left( 1 + 2 \sqrt{\frac{2 \log \left( \frac{2(k+1)}{p} \right)}{d}} \right) ||g||^2$$

with probability at most $\frac{p}{k+1}$.

Finally, by plugging the above bound and the bound (13) into the bound (12), we obtain that

$$\langle \hat{x}_{LSQ}, \boldsymbol{g} \rangle - \langle \hat{x}_{NES}, \boldsymbol{g} \rangle \leq \left( \sqrt{\frac{32k \log(2(k+1)/p)}{d}} \right) \cdot 2 \log \left( \frac{2(k+1)}{p} \right) \left( 1 + 2 \sqrt{\frac{2 \log \left( \frac{2(k+1)}{p} \right)}{d}} \right) ||g||^2$$

$$\leq 8 \sqrt{\frac{2k}{d} \cdot \log^3 \left( \frac{2k+2}{p} \right)} \left( 1 + \frac{\kappa}{\sqrt{d}} \right) ||g||^2,$$

with probability $1 - p$, where

$$\kappa = 2 \sqrt{\log \left( \frac{2k(k+1)}{p} \right)}.$$

This completes the proof. $\qquad\square$

**Theorem 2** (Least-Squares Optimality). *For a fixed projection matrix $A$ and under the following observation model of isotropic Gaussian noise: $\boldsymbol{y} = A\boldsymbol{g} + \vec{\varepsilon}$ where $\boldsymbol{\varepsilon} \sim \mathcal{N}(\boldsymbol{0}, \varepsilon\boldsymbol{Id})$, the least-squares estimator as in Theorem 1, $\hat{x}_{LSQ} = A^T(AA^T)^{-1}\boldsymbol{y}$ is a finite-sample efficient (minimum-variance unbiased) estimator of the parameter $\boldsymbol{g}$.*

*Proof.* Proving the theorem requires an application of the Cramer-Rao Lower Bound theorem:

**Theorem 3** (Cramer-Rao Lower Bound). *Given a parameter $\theta$, an observation distribution $p(x; \theta)$, and an unbiased estimator $\hat{\theta}$ that uses only samples from $p(x; \theta)$, then (subject to Fisher regularity conditions trivially satisfied by Gaussian distributions),*

$$Cov\left[\hat{\theta} - \theta\right] = \mathbb{E}\left[(\hat{\theta} - \theta)(\hat{\theta} - \theta)^T\right] \geq [I(\theta)]^{-1} \text{ where } I(\theta) \text{ is the Fisher matrix: } [I(\theta)]_{ij} = -\mathbb{E}\left[\frac{\partial \log p(x; \theta)}{\partial \theta_i \partial \theta_j}\right]$$

Now, note that the Cramer-Rao bound implies that if the variance of the estimator $\hat{\theta}$ is the inverse of the Fisher matrix, $\hat{\theta}$ must be the minimum-variance unbiased estimator. Recall the following form of the Fisher matrix:

$$I(\theta) = \mathbb{E}\left[\left(\frac{\partial \log p(x; \theta)}{\partial \theta}\right)\left(\frac{\partial \log p(x; \theta)}{\partial \theta}\right)^T\right] \tag{14}$$

Now, suppose we had the following equality, which we can then simplify using the preceding equation:

$$I(\theta)\left(\hat{\theta} - \theta\right) = \frac{\partial \log p(x; \theta)}{\partial \theta} \tag{15}$$

$$\left(I(\theta)\left(\hat{\theta} - \theta\right)\right)\left(I(\theta)\left(\hat{\theta} - \theta\right)\right)^T = \left(\frac{\partial \log p(x; \theta)}{\partial \theta}\right)\left(\frac{\partial \log p(x; \theta)}{\partial \theta}\right)^T \tag{16}$$

$$\mathbb{E}\left[\left(I(\theta)\left(\hat{\theta} - \theta\right)\right)\left(I(\theta)\left(\hat{\theta} - \theta\right)\right)^T\right] = \mathbb{E}\left[\left(\frac{\partial \log p(x; \theta)}{\partial \theta}\right)\left(\frac{\partial \log p(x; \theta)}{\partial \theta}\right)^T\right] \tag{17}$$

$$I(\theta)\mathbb{E}\left[(\hat{\theta} - \theta)(\hat{\theta} - \theta)^T\right]I(\theta) = I(\theta) \tag{18}$$

Multiplying the preceding by $[I(\theta)]^{-1}$ on both the left and right sides yields:

$$\mathbb{E}\left[(\hat{\theta} - \theta)(\hat{\theta} - \theta)^T\right] = [I(\theta)]^{-1}, \tag{19}$$

which tells us that (15) is a sufficient condition for finite-sample efficiency (minimal variance). We show that this condition is satisfied in our case, where we have $y \sim A\boldsymbol{g} + \varepsilon$, $\hat{\theta} = \hat{x}_{LSQ}$, and $\theta = \boldsymbol{g}$. We begin by computing the Fisher matrix directly, starting from the distribution of the samples $y$:

$$p(y; \boldsymbol{g}) = \frac{1}{\sqrt{(2\pi\varepsilon)^d}} \exp\left\{\frac{1}{2\varepsilon}(y - A\boldsymbol{g})^T(y - A\boldsymbol{g})\right\} \tag{20}$$

$$\log p(y; \boldsymbol{g}) = \frac{d}{2}\log(2\pi\varepsilon) + \frac{1}{2\varepsilon}(y - A\boldsymbol{g})^T(y - A\boldsymbol{g}) \tag{21}$$

$$\frac{\partial \log p(y; \boldsymbol{g})}{\partial \boldsymbol{g}} = \frac{1}{2\varepsilon}\left(2A^T(y - A\boldsymbol{g})\right) \tag{22}$$

$$= \frac{1}{\varepsilon}A^T(y - A\boldsymbol{g}) \tag{23}$$

$$\tag{24}$$

Using (14),

$$I(\boldsymbol{g}) = \mathbb{E}\left[\left(\frac{1}{\varepsilon}A^T(y - A\boldsymbol{g})\right)\left(\frac{1}{\varepsilon}A^T(y - A\boldsymbol{g})\right)^T\right] \tag{25}$$

$$= \frac{1}{\varepsilon^2}A^T\mathbb{E}\left[(y - A\boldsymbol{g})(y - A\boldsymbol{g})^T\right]A \tag{26}$$

$$= \frac{1}{\varepsilon^2} A^T (\varepsilon \boldsymbol{Id}) A \tag{27}$$

$$= \frac{1}{\varepsilon} A^T A \tag{28}$$

Finally, note that we can write:

$$I(\boldsymbol{g})(\hat{x}_{LSQ} - \boldsymbol{g}) = \frac{1}{\varepsilon} A^T A (A^T (AA^T)^{-1} y - \boldsymbol{g}) \tag{29}$$

$$= \frac{1}{\varepsilon} (A^T y - A^T A \boldsymbol{g}) \tag{30}$$

$$= \frac{\partial \log p(y; \boldsymbol{g})}{\partial \boldsymbol{g}}, \tag{31}$$

which concludes the proof, as we have shown that $\hat{x}_{LSQ}$ satisfies the condition (15), which in turn implies finite-sample efficiency. $\square$

**Claim 1.** *Applying the precise bound that we can derive from Theorem 1 on an ImageNet-sized dataset $(d = 300000)$ and using $k = 100$ queries (what we use in our $\ell_\infty$ threat model and ten times that used for our $\ell_2$ threat model),*

$$\langle \hat{x}_{LSQ}, \boldsymbol{g} \rangle - \langle \hat{x}_{NES}, \boldsymbol{g} \rangle \leq \frac{5}{4} ||g||^2.$$

*For 10 queries,*

$$\langle \hat{x}_{LSQ}, \boldsymbol{g} \rangle - \langle \hat{x}_{NES}, \boldsymbol{g} \rangle \leq \frac{1}{2} ||g||^2.$$

# B    OMITTED FIGURES

## B.1    COMPRESSIVE SENSING

Compressed sensing approaches can, in some cases, solve the optimization problem presented in Section 2.4. However, these approaches require sparsity to improve over the least squares method. Here we show the lack of sparsity in gradients through a classifier on a set of canonical bases for images. In Figure 5, we plot the fraction of $\ell_2$ weight accounted for by the largest $k$ components in randomly chosen image gradients when using two canonical bases: standard and wavelet (db4). While lack of sparsity in these bases does not strictly preclude the existence of a basis on which gradients are sparse, it suggests the lack of a fundamental structural sparsity in gradients through a convolutional neural network.

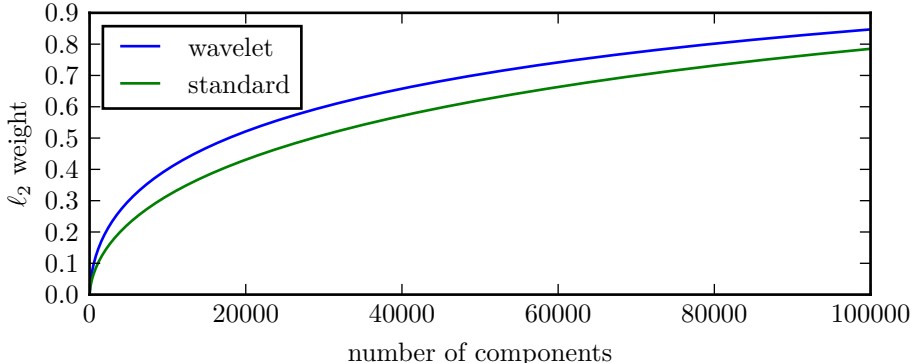

Figure 5: Sparsity in standard, wavelet (db4 wavelets), and PCA-constructed bases for the gradients of 5,000 randomly chosen example images in the ImageNet validation set. The y-axis shows the mean fraction of $\ell_2$ weight held by the largest $k$ vectors over the set of 5,000 chosen images. The x-axis varies $k$. The gradients are taken through a standardly trained Inception v3 network. None of the bases explored induce significant sparsity.

## B.2    TILING

An example of the tiling procedure applied to a gradient can be seen in Figure 6.

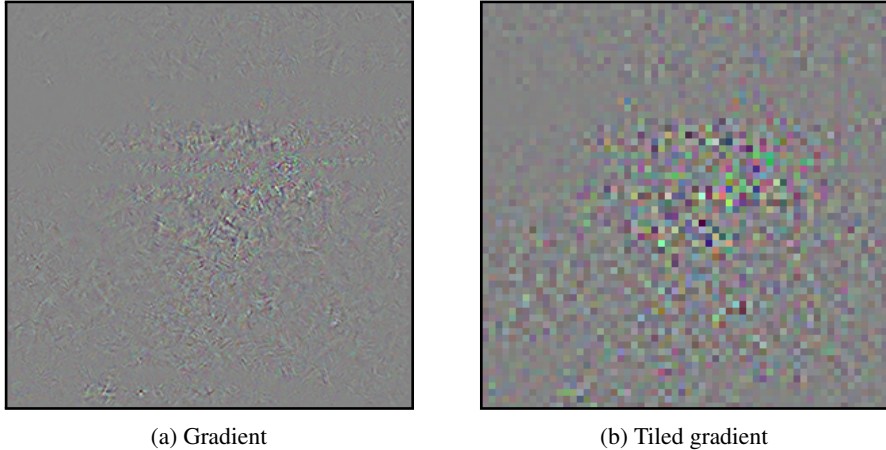

| (a) Gradient | (b) Tiled gradient |

Figure 6: Average blurred gradient with kernel size or "tile length" 5. The original gradient can be seen in 6a, and the "tiled" or average blurred gradient can be seen in 6b

### B.3 TIME-DEPENDENT PRIORS AT HIGHER STEP SIZES

We show in Figure 7 that the correlation between successive gradients on the NES trajectory are signficantly correlated, even at much higher step sizes (up to $\ell_2$ norm of 4.0, which is a typical value for $\varepsilon$, the total adversarial perturbation bound and thus an absolute bound on step size). This serves as further motivation for the time-dependent prior.

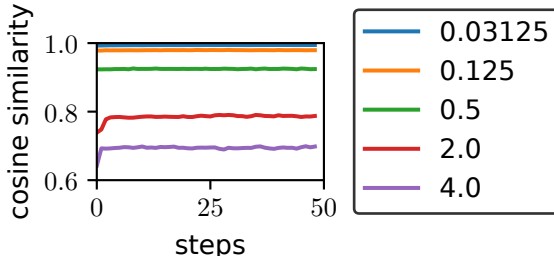

Figure 7: Figure 2 repeated for several step sizes, showing that the successive correlation between gradients continues even at higher step sizes.

## C  HYPERPARAMETERS

Table 2: Hyperparameters for the NES approach.

| Hyperparameter | Value | | |
| --- | --- | --- | --- |
| | ImageNet $\ell_\infty$ | ImageNet $\ell_2$ | CIFAR10 $\ell_\infty$ |
| Samples per step | 100 | 10 | 50 |
| Learning Rate | 0.01 | 0.3 | 0.01 |

Table 3: Hyperparameters for the bandits approach (variables names as used in pseudocode).

| Hyperparameter | Value | | |
| --- | --- | --- | --- |
| | ImageNet $\ell_\infty$ | ImageNet $\ell_2$ | CIFAR10 $\ell_\infty$ |
| $\eta$ (OCO learning rate) | 100 | 0.1 | 100 |
| $h$ (Image $\ell_p$ learning rate) | 0.005 | 0.5 | 0.0001 |
| $\delta$ (Bandit exploration) | 0.01 | 0.01 | 0.01 |
| $\eta$ (Finite difference probe) | 0.01 | 0.01 | 0.01 |
| Tile size (Data-dependent prior only) | $(6px)^2$ | $(6px)^2$ | $(10px)^2$ |

Table 4: Experimental setup for comparing Bandits-NES. Setup and results for comparison with Tu et al. (2018) in Appendix G

| Parameter | Value | | |
| --- | --- | --- | --- |
| | ImageNet $\ell_\infty$ | ImageNet $\ell_2$ | CIFAR10 $\ell_\infty$ |
| Max allowed queries | | 10,000 | |
| Test set size | 10,000 | 10,000 | 1,000 |
| Allowed perturbation $\varepsilon$ | 0.05 | 5.0 | 0.05 |

# D FULL RESULTS

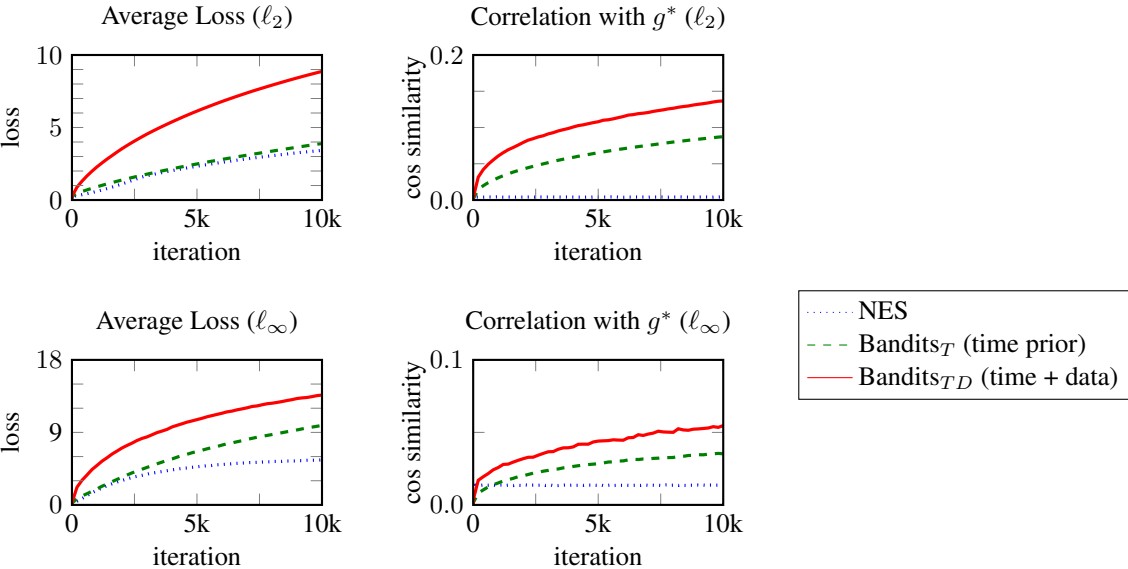

Figure 8: *Average loss and cosine distance versus number of queries used over the approaches' optimization trajectories in the two threat models. We average each cosine distance and loss point at each query number over 100 images from the evaluation set.*

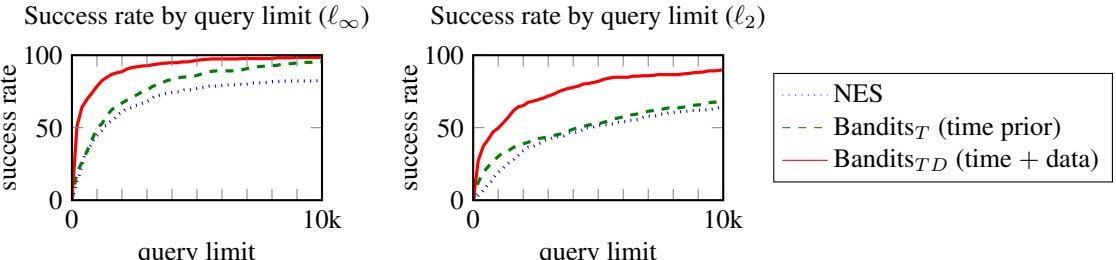

Figure 9: *Cumulative distribution functions for the number of queries required to create an adversarial example in the $\ell_2$ and $\ell_\infty$ settings for the NES, bandits with time prior (Bandits$_T$), and bandits with time and data-dependent priors (Bandits$_{TD}$) approaches. Note that the CDFs do not converge to one, as the approaches sometimes cannot find an adversarial example in less than 10,000 queries.*

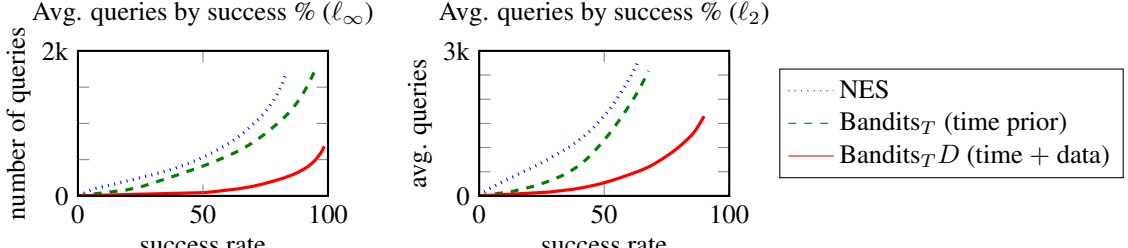

Figure 10: *The average number of queries used per successful image for each method when reaching a specified success rate: we compare NES Ilyas et al. (2017), Bandits$_T$ (our method with time prior only), and Bandits$_{TD}$ (our method with both data and time priors) and find that our methods strictly dominate NES—that is, for any desired sucess rate, our methods take strictly less queries per successful image than NES.*

# E   RESULTS FOR CIFAR-10

Here, we give results for the CIFAR-10 dataset, comparing our best method (Bandits$_{TD}$) and NES. We train Inception-v3, ResNet-50, and VGG16 classifiers by fine-tuning the standard PyTorch ImageNet classifiers. As such, all images are upsampled to $224 \times 224$ ($299 \times 299$) for ResNet-50 and VGG16 (and Inception-v3). Just as for ImageNet, we use a maximum $\ell_\infty$ perturbation of $0.05$, where images are scaled to $[0, 1]$.

Table 5: Summary of effectiveness of $\ell_\infty$ CIFAR10 attacks on Inception v3, ResNet-50, and VGG16 (I, R, V) using NES and bandits with time and data-dependent priors (Bandits$_{TD}$). Note that in the first column, the average number of queries is calculated only over successful attacks, and we enforce a query limit of 10,000 queries. For purposes of direct comparison, the last column calculates the average number of queries used for only the images that NES (previous SOTA) was successful on. Our most powerful attack uses 2-4 times fewer queries, and fails 2-22 times less often.

| Attack | Avg. Queries | | | Failure Rate | | | Queries on NES Success | | |
|---|---|---|---|---|---|---|---|---|---|
| | I | R | V | I | R | V | I | R | V |
| NES | 1202 | 1317 | 879 | 22% | 31% | 27% | 1202 | 1317 | 879 |
| **Bandits$_{TD}$** | **602** | **554** | **509** | **0.6%** | **12%** | **18%** | **439** | **399** | **388** |

# F    RESULTS FOR OTHER CLASSIFIERS

Here, we give results for the ImageNet dataset, comparing our best method (Bandits$_{TD}$) and NES for Inception-v3 (also shown in Table 1), VGG16, and ResNet50 classifiers. Note that we do not fine-tune the hyperparameters to the new classifiers, but simply use the hyperparameters found for Inception-v3. Nevertheless, our best method consistently outperforms NES on black-box attacks.

Table 6: Summary of effectiveness of $\ell_\infty$ and $\ell_2$ ImageNet attacks on Inception v3, ResNet-50, and VGG16 (I, R, V) using NES and bandits with time and data-dependent priors (Bandits$_{TD}$). Note that in the first column, the average number of queries is calculated only over successful attacks, and we enforce a query limit of 10,000 queries. For purposes of direct comparison, the last column calculates the average number of queries used for only the images that NES (previous SOTA) was successful on. Our most powerful attack uses 2-4 times fewer queries, and fails 2-5 times less often.

| Attack | | Avg. Queries | | | Failure Rate | | | #Q on NES Success | | |
|---|---|---|---|---|---|---|---|---|---|---|
| | | I | R | V | I | R | V | I | R | V |
| $\ell_2$ | NES | 2938 | 2193 | 1244 | 34.4% | 10.1% | **11.6%** | 2938 | 2193 | 1244 |
| | **Bandits$_{TD}$** | **1858** | **993** | **594** | **15.5%** | **9.7%** | 17.2% | **999** | **1195** | **1219** |
| $\ell_\infty$ | NES | 1735 | 1397 | 764 | 22.2% | 10.4% | 10.5% | 1735 | 1397 | 764 |
| | **Bandits$_{TD}$** | **1117** | **722** | **370** | **4.6%** | **3.4%** | **8.4%** | **703** | **594** | **339** |

## G    COMPARISON TO (TU ET AL, 2018)

To compare with the method of Tu et al. (2018), we consider the same classifier and dataset (Inception-v3 and Imagenet) under the same $\ell_2$ threat model. Note that Tu et al. (2018) use mean rather than maximum $\ell_2$ perturbation to evaluate their attacks (since the method is based on a Lagrangian relaxation). To ensure a fair comparison we compare against the average number of queries to reach the adversarial examples bounded within a pertubation budget of $2 \cdot 10^{-4}$, which is explicitly reported byTu et al. (2018).

For the bandits approach, we used Bandits$_T$, (the bandits method with the time prior) and Bandits$_{TD}$ (the bandits method with both time and data prior) and run the methods until 100% success is reached. We use the same hyperparameters from the *untargeted* ImageNet experiments (given in Appendix C). Our findings, given in Table 7 show that our best method achieves an 100% success rate, and an over 6-fold reduction in queries. Note that the method of Tu et al. (2018) achieves 100% success rate in general, but only constrains the mean $\ell_2$ perturbation, and thus actually achieves a strictly less than 100% success rate with this perturbation threshold.

Table 7: Comparison against coordinate-based query efficient finite differences attacks from Tu et al. (2018), using the ImageNet dataset, with a maximum $\ell_2$ constraint of 0.0002 per-pixel normalized (which is equal to a max-$\ell_2$ threshold reported by Tu et al. (2018)). For our methods (Bandits$_T$ and Bandits$_{TD}$) we use the same hyperparameters as in our comparison to NES, which are given in Appendix C.

| Attack | Avg. Queries | Success Rate |
|---|---|---|
| AutoZOOM-BiLin (Tu et al., 2018) | 15,064 | <100% |
| AutoZOOM-AE (Tu et al., 2018) | 14,914 | <100% |
| Bandits$_T$ (Ours) | 4455 | **100%** |
| **Bandits$_{TD}$ (Ours)** | **2297** | **100%** |

