# OpenReview forum: "Prior Convictions: Black-box Adversarial Attacks with Bandits and Priors"
_ICLR.cc/2019/Conference_

### Official Review · AnonReviewer2 · 2018-11-02
**A decent paper**

**Rating:** 7
**Confidence:** 3

**Review:**

UPDATE:

I've read the revised version of this paper, I think the concernings have been clarified.

-------

This paper proposes to employ the bandit optimization based approach for the generation of adversarial examples under the loss accessible black-box situation. The authors examine the feasibility of using the step and spatial dependence of the image gradients as the prior information for the estimation of true gradients. The experimental results show that the proposed method out-performs the Natural evolution strategies method with a large margin.

Although I think this paper is a decent paper that deserves an acceptance, there are several concernings:

1. Since the bound given in Theorem 1 is related to the square root of k/d, I wonder if the right-hand side could become "vanishingly small", in the case such as k=10000 and d=268203. I wish the authors could explain more about the significance of this Theorem, or provide numerical results (which could be hard).

2. Indeed I am not sure if Section 2.4 is closely related to the main topic of this paper, these theoretical results seem to be not helpful in convincing the readers about the idea of gradient priors. Also, the length of the paper is one of the reasons for the rating.

3. In the experimental results, what is the difference between one "query" and one "iteration"? It looks like in one iteration, the Algorithm 2 queries twice?

---

> ### Author Response · Authors · 2018-11-13
> **Response**
>
> We thank the reviewer for the detailed comments on the paper. We address the main points below:
>
> 1. Typically black-box adversarial attacks are executed in a multi-step fashion, i.e. by using small numbers of queries per gradient estimates, and taking several gradient estimate steps (Ilyas et al, the NES-based attack, for example, uses 50 queries per gradient estimate). While it may be possible to prove tighter bounds, in the 50-query regime with d=268203, the bound is actually rather tight. (Furthermore, during our own preliminary experimentation, least-squares attacks usually performed identically to NES).
>
> 2. Section 2.4 is meant to illustrate that without priors, we have essentially hit the limit of query-efficiency in black-box attacks. In particular, NES, which we found to be the current state-of-the-art, actually turns out to be approximately optimal, even from a theoretical perspective. This motivates us to take a new look on adversarial example generation, breaking through this optimality by introducing new information into the problem.
>
> Without the proof in Section 2.4, one could reasonably hope that there are simply better gradient estimators that we can use as a drop-in replacement for NES. The theorems we prove there instead motivate our bandit optimization-based view.
>
> 3. One iteration constitutes two queries (which are used for a variance-reduced gradient estimate via antithetic sampling). In general, the query count refers to queries of the classifier, whereas iteration counts the number of times that we take an estimated gradient step.
>
> We hope the above points clarify the reviewer's concerns, and thank the reviewer again for the detailed feedback.

---

> ### Author Response · Authors · 2018-11-25
> **Revision [Reply to R2]**
>
> Thank you again for the review. We have now posted a revision of our paper, and the summary comment above details all of the changes we've made in response to reviewer comments, including several additional experiments and comparisons with other methods.
>
> To highlight the changes that are most relevant to your review:
>
> 1) We now provide an illustration of the bound in Appendix A in the relevant query regimes
>
> 2 and 3) We have clarified some points in the papers based on reviewer comments and added significantly more experimental results---we hope that these results further justify the use of the full 10 pages.

---

### Official Review · AnonReviewer3 · 2018-11-03
**Good paper, low confidence.**

**Rating:** 8
**Confidence:** 2

**Review:**

Paper formalizes the gradient estimation problem in a black-box setting, and provs the equivalence of least Squares with NES. It then improves on state of the art by using priors coupled with a bandit optimization technique.

The paper is well written. The idea of using priors to improve adversarial gradient attacks is an enticing idea. The results seem convincing.

Comments:
- I missed how data dependent prior is factored into the algorithms 1-3. Is it by the choice of d? I suggest a clearer explanation.
- In fig 4, I was confused that the loss of the methods is increasing. it took me a minute to realize this is the maximized adversarial loss, and thus higher is better. you may want to spell this out for clarity. I typically associate lower loss with better algorithms.
- I am confused by Fig 4c. If I am comparing g to g*, I do expect a high cosine similarity. cos = 1 is the best. Why is correlation so small? and why is it 0 for NES? You may also want to offer additional insight in the text explaining 4c.

Minor comments:
- Is table one misplaced?
- The symbol for "boundary of set U" may be confused with a partial derivative symbol
- first paragraph of 2.4: "our estimator a sufficiently". something missing?
- "It is the actions g_t (equal to v_t) which..." refering to g_t as actions is confusing. Although may be technically correct in bandit setting
- Further explain the need for the projection of algorithm 3, line 7.
- Fig 4: refer to true gradient as g*

Caveat: Although I am well versed in bandits, I am not familiar with adversarial training and neural network literature. There is a chance I may have misevaluated central concepts of the paper.

---

> ### Author Response · Authors · 2018-11-13
> **Response**
>
> We thank the reviewer for the comments!
>
> We address the main points below:
>
> > Data dependent prior in pseudocode: Yes it is in fact by choice of d, but we agree this can be made clearer in the pseudocode. We will make sure to describe this more clearly in our final paper.
>
> > Figure 4: We will make sure to update this and be more explicit.
>
> > Figure 4c (low cosine similarity): Remarkably, for black-box attacks, though higher cosine similarity is better, the threshold for a successful adversarial attack (in terms of cosine similarity) is extremely low. In particular, for NES, the cosine similarity (as you mentioned) is almost 0, but the gradient estimates *still* result in a successful attack! We show that using our method leads to significantly better estimates of the gradient, though as one would expect in such a query-deficient domain (100s of queries vs 3*10e5 dimensional images), still pretty poor.
>
> We will also be sure to address all of the minor comments in our final paper. We thank the reviewer again for the useful comments and suggestions.

---

> ### Author Response · Authors · 2018-11-25
> **Revision [Reply to R3]**
>
> We have addressed the above comments in our revision, please see the main comment for more details. Thank you again for the review and suggestions.

---

### Official Review · AnonReviewer1 · 2018-11-06
**good paper, accept**

**Rating:** 7
**Confidence:** 5

**Review:**

This paper formulates the black-box adversarial attack as a gradient estimation
problem, and provide some theoretical analysis to show the optimality of an
existing gradient estimation method (Neural Evolution Strategies) for black-box
attacks.

This paper also proposes two additional methods to reduce the number of queries
in black-box attack, by exploiting the spacial and temporal correlations in
gradients. They consider these techniques as priors to gradients, and a bandit
optimization based method is proposed to update these priors.

The ideas used in this paper are not entirely new. For example, the main
gradient estimation method is the same as NES (Ilyas et al. 2017);
data-dependent priors using spatially local similarities was used in Chen et
al. 2017.  But I have no concern with this and the nice thing of this paper is
to put these tricks under an unified theoretical framework, which I really
appreciate.

Experiments on black-box attacks to Inception-v3 model show that the proposed
bandit based attack can significantly reduces the number of queries (2-4 times
fewer) when compared with NES.

Overall, the paper is well written and ideas are well presented.
I have a few concerns:

1) In Figure 2, the authors show that there are strong correlations between the
gradients of current and previous steps. Such correlation heavily depends on
the selection of step size.  Imagine that the step size is sufficiently large,
such that when we arrive at a new point for the next iteration, the
optimization landscape is sufficiently changed and the new gradient is vastly
different than the previous one. On the other hand, when using a very small
step-size close to 0, gradients between consecutive steps will be almost the
same. By changing step-size I can show any degree of correlation.  I am not
sure if the improvement of Bandit_T comes from a specific selection of
step-size. More empirical evidence on this need to be shown - for example, run
Bandit_T and NES with different step sizes and observe the number of queries
required.

2) This paper did not compare with many other recent works which claim to
reduce query numbers significantly in black-box attack. For example, [1]
proposes "random feature grouping" and use PCA for reducing queries, and [2]
uses a good gradient estimator with autoencoder. I believe the proposed method
can beat them, but the authors should include at least one more baseline to
convince the readers that the proposed method is indeed a state-of-the-art.

3) Additionally, the results in this paper are only shown on a single model
(Inception-v3), and it is hard to compare the results directly with many other
recent works. I suggest adding at least two more models for comparison (most
black-box attack papers also include MNIST and CIFAR, which should be easy to
add quickly). These numbers can be put in appendix.

Overall, this is a great paper, offering good insights on black-box adversarial
attack and provide some interesting theoretical analysis. However currently it
is still missing some important experimental results as mentioned above, and
not ready to be published as a high quality conference paper. I conditionally
accept this paper as long as sufficient experiments can be added during the
discussion period.


[1] Exploring the Space of Black-box Attacks on Deep Neural Networks, by Arjun
Nitin Bhagoji, Warren He, Bo Li and Dawn Song, https://arxiv.org/abs/1712.09491
(conference version accepted by ECCV 2018)

[2] AutoZOOM: Autoencoder-based Zeroth Order Optimization Method for Attacking
Black-box Neural Networks, by Chun-Chen Tu, Paishun Ting, Pin-Yu Chen, Sijia
Liu, Huan Zhang, Jinfeng Yi, Cho-Jui Hsieh, Shin-Ming Cheng,
https://arxiv.org/abs/1805.11770

==========================================

After discussing with the authors, they provided better evidence to support the conclusions in this paper, and fixed bugs in experiments. The paper looks much better than before. Thus I increased my rating.

---

> ### Author Response · Authors · 2018-11-13
> **Response**
>
> Thank you for the detailed comments, we will be sure to make these changes in the final version of the paper. As the reviewer correctly identifies, we consider the theoretical framework of online optimization as a basis for all black-box attacks to be one of our most profound contributions. That said, in order to improve the quality of the experimental results, we have addressed and added each suggested experiment. Specifically:
>
> 1) We thank the reviewer for raising this---we initially only used the default NES step size (from Ilyas et al) to evaluate the temporal correlation. To give a fuller picture on how this temporal correlation relates with the step size, we have added a new plot in the appendix, which shows the average correlation on a trajectory as a function of the step size.
>
> 2) To address this, we have added a table (in the Appendix) which compares our query-efficiency against that of [1] and [2]. It should also be noted, however, that both [1] and [2] can be integrated as "priors" on the gradient; in particular, that the gradient lays in some low-dimensional subspace. Our framework gives us a way to formalize these assumptions, and measure how empirically valid they are in order to find better and better black-box attacks.
>
> 3) We have also added results on ResNet-50 and VGG-16 on ImageNet, and have also benchmarked our attack on all three classifiers (Inceptionv3, ResNet-50, VGG-16) on CIFAR as well.
>
> We will be sure to comment again with a revision when the experiments are complete and integrated into the paper. We thank the reviewer again for the valuable suggestions.

---

> > ### Comment · AnonReviewer1 · 2018-11-27
> > **Thanks for adding these results! They look very good, except for a small concern**
> >
> > Dear Paper1206  Authors,
> >
> > Thank you for adding these new results. Figure 7 now shows the cosine similarity under different step sizes, which looks convincing. The newly added experiments on different models (ResNet-50, VGG-16) and different datasets (CIFAR and ImageNet), as well as the comparisons to other state-of-the-art methods make this paper look much stronger than before.
> >
> > I have a concern regarding the comparison with (Tu et al, 2018). The 100-fold reduction looks to good to be true. Can you confirm that you performed the attack under the same setting? e.g., do you run attacks with the same target labels for both methods, or running untargeted attacks for both? I think it is better to double check this.
> >
> > I am willing to increase my rating to 7 as long as the above concern can be addressed.
> >
> > Thanks,
> > Paper1206 AnonReviewer1

---

> > > ### Author Response · Authors · 2018-11-27
> > > **Thank you for pointing this out... have revised the paper and updated the result**
> > >
> > > First of all, we would like to sincerely thank the reviewer for their continually detailed comments and thorough review---it has been great help in improving the manuscript.
> > >
> > > Upon checking the code, we realized that (as the reviewer suggested), we had accidentally reproduced the _targeted_ attacks in the baseline code repository. To account for this, we modified our code to work for targeted attacks, and properly replicated the experimental setup, choosing the correct \ell_2 perturbation bound, and random target classes as in Tu et al (except for the fact that we use the prepackaged Inception-v3 classifier rather than the downloaded one from Tu et al). We don't tune our hyper parameters at all, and use the same ones that we used for untargeted.
> > >
> > > Our method achieves the same success rate at over 3 times the query efficiency at 100% success rate (note that this is higher success rate than Tu et al achieve at the same l2 perturbation bound, since there the authors only bound the mean and not the max), still establishing significant improvement. We have uploaded a revision reflecting these changes.

---

> > > > ### Author Response · Authors · 2018-11-27
> > > > **Another note**
> > > >
> > > > Also note that due to the time constraint in getting a revision in, we actually only compared Tu et al to our second-best method, Bandits_T. We are running a comparison with Bandits_{TD} (both time and data priors) and will revise (if time permits) and/or report back the results.
> > > >
> > > > [EDIT: we have now done so, see comment above]

---

> > > > ### Author Response · Authors · 2018-11-27
> > > > **Updated again with time and data priors**
> > > >
> > > > We have updated the paper again (specifically, the comparison with Tu et al) to reflect experiments we have now run with both the time and data prior (Bandits_{TD}). At 100% success rate with the same experimental design, our method now uses over 6 times fewer queries.

---

> > > > > ### Comment · AnonReviewer1 · 2018-11-27
> > > > > **Thanks for the update. I will increase my rating to 7**
> > > > >
> > > > > Thanks for fixing this bug in experiments. The results look much reasonable now. I will increase my rating.

---

> ### Author Response · Authors · 2018-11-25
> **Revision [Reply to R1]**
>
> We have addressed comments (1), (2), and (3) in our revision (details are in the main comment above). To address the raised points directly:
>
> (1) is now addressed in Figure 7 in Appendix B.3 which shows how the time-dependent trend decays with the step size---even at high step sizes the trend persists. Specifically, we plot a graph identical to Figure 2 but for many different step sizes, from norm around 0.03 all the way to 4.0.
>
> (2) Appendix G now shows a comparison with Tu et al ([2] in the original review). See our main comment above for a summary of the results.
>
> (3) We now include results from ImageNet and CIFAR, with Inception-v3, ResNet, and VGG16 in the appendices (more details in our main comment above).
>
> Thank you again for the detailed review and the useful suggestions.

---

### Author Response · Authors · 2018-11-25
**Revision**

We thank all the reviewers again for the helpful responses and revision suggestions. We have posted a revision that we believe addresses all the reviewer comments.

In addition to adding the suggested edits to the paper for clarity, we have now compared our approach with several datasets, baselines, and classifiers, and established a significant margin over state-of-the-art methods. Specifically, we have made the following updates:

—————
Quantifying time-dependent prior
—————
We include a graph (in the omitted figures appendix) showing that the successive correlation prior (aka the time-dependent prior) holds true even up to very large step sizes. Specifically, we plot a graph identical to Figure 2 but for many different step sizes, from norm around 0.03 all the way to 4.0.

—————
Other threat models and datasets
—————
We have added an Appendix F corresponding to ImageNet results for VGG16 and ResNet50 classifiers (along with Inceptionv3 copied from the main text for reference). Our methods still outperform NES on these benchmarks, often by a larger margin than shown for Inception-v3 in Table 1.

We have added an Appendix E corresponding to a comparison of our methods and NES in the CIFAR l-infinity threat model (for L2, we could not find a reasonable maximum \epsilon from recent literature) with VGG16, Resnet50, and Inceptionv3 networks.

————-
Comparison with another baseline
————-

Efficiency compared to Tu et al:
—————
We looked into (Tu et al, 2018) and (Bhagoji et al, 2017) as suggested by reviewer 1 to compare with a baseline; we chose to compare with Tu et al (AutoZOOM) since it was (a) released later, (b) uses a more standard classifier than in Bhagoji et al and (c) does not require access to an external set of representative images (unlike Bhagoji et al, which uses this set to find the PCA components). As such, we have added an Appendix comparing our method to that of Tu et al: achieving the same success rate and using the mean perturbation from Tu et al as our maximum perturbation, we achieve a 35-fold reduction in query complexity.

Efficiency compared to Tu et al + fine tuning:
—————-
 Tu et al also give a “distortion fine-tuning” technique that attempts to reduce the mean perturbation after the attack. This fine-tuning takes around 100,000 queries, and in the best case, after using around 100,000 queries, reduces the mean perturbation to 0.4e-4 per-pixel normalized, which works out to just over 10 (see Figure 3a in Tu et al). In Appendix F, we show that running our attack with this lower distortion budget directly gives a similar success rate, using an average of around 900 queries as opposed to 100,000, giving more than a *100-fold* reduction in query complexity.

————
Bound illustration
————
- To illustrate, we give an example of our own \ell_2 threat model, where Theorem 1 gives us a bound on the performance gap between NES and least squares, in Appendix 1 (after the proofs).

————
Edits to paper
————
- We noticed that our image normalization for generating Table 1 was slightly incorrect, so we have fixed it and rerun the experiment—this has not changed the output significantly, and our methods still beat NES by the same margin of normalized queries. However, in the interest of correctness, we have updated the numbers in Table 1 to reflect the experiment run with correct normalization.
- We have made the pseudocode for the bandits attack clearer, and explicitly noted how the data-dependent prior can be included, as well as justifying the boundary projection step
- Fixed: \nabla L —> g^* in Figures
- Fixed: Section 2.4 sentence (as pointed out by Reviewer 3)

---

### Meta-Review · Area_Chair1 · 2018-12-17
**well written, effective and relevant work on blackbox adversarial example generation**

**Confidence:** 4
**Recommendation:** Accept (Poster)

**Metareview:**

This paper is on the problem of adversarial example generation in the setting where the predictor is only accessible via function evaluations with no gradients available. The associated problem can be cast as a blackbox optimization problem wherein finite difference and related gradient estimation techniques can be used. This setting appears to be pervasive. The reviewers agree that the paper is well written and the proposed bandit optimization-based algorithm provides a nice framework in which to integrate priors, resulting in impressive empirical improvements.